# Investigation of a Calibration Method of Coriolis Mass Flowmeters by Density- and Pressure-Matching Approaches for Hydrogen Refueling Stations

**Woong Kang** [1]**, Jinwoo Shin** [1]**, Byungro Yoon** [1]**, Sunghee Kil** [2]**, Sangsik Yim** [2]**, Wonguk Han** [2] **and Unbong Baek** [1,*]

1   Korea Research Institute of Standard and Science, 267 Gajeong-ro, Yuseong-gu,
    Daejeon 34113, Republic of Korea
2   Korea Gas Safety Corporation, 1390 Wonjung-ro, Eumseong 27738, Republic of Korea
*   Correspondence: ubbaek@kriss.re.kr; Tel.: +82-042-868-5384

**Abstract:** Hydrogen fuel cell electric vehicles are emerging as a means of transportation using renewable and carbon-free energy due to global warming and air pollution. Hydrogen fuel cell electric vehicles are typically refueled at a wide range of temperatures ($-40\ ^\circ$C to 85 $^\circ$C) in hydrogen refueling stations in accordance with globally accepted standards. Currently, there is no traceable method by which to verify and calibrate the Coriolis mass flowmeters used at hydrogen refueling stations, except for a water calibration process as a conventional method for mass flowrate calibration. To verify the hydrogen flow metering to a suitable level of accuracy under the challenging condition of high pressures and a wide range of temperatures, necessary methodologies and calibration facilities are developed in the present study. A flow measurement characteristic test of the hydrogen mass flowmeter under identical density conditions of the refueled hydrogen was conducted using the high-pressure gas flow standard system of the Korea Research Institute of Standards and Science to assess the effects on the medium and pressure of the mass flowmeter in a density-matching approach. To investigate the pressure dependence of the mass flowmeter at a hydrogen refueling station, a high-pressure water flow test was conducted in the pressure range of 2 bar to 700 bar, which is a pressure-matching approach. Finally, the KRISS Hydrogen Field Test Standard based on the gravimetric principle was developed to verify the measurement accuracy of the mass flowmeter to be used at hydrogen refueling stations for the first time in Korea.

**Keywords:** hydrogen refueling station; hydrogen flow metering; Coriolis mass flowmeter

## 1. Introduction

As interest in developing eco-friendly cars has grown due to global warming and air pollution, hydrogen fuel cell electric vehicles are emerging as a measure of renewable and carbon-free energy in the world. Hydrogen vehicles are currently under development and are commercially available from several manufacturers, mostly in South Korea and Japan. In connection with the supply and distribution of hydrogen vehicles, there is increasing national and policy support for the development of the hydrogen infrastructure required, specifically hydrogen refueling stations, in many countries. To usher in the hydrogen economy and future energy transition, South Korea promulgated the Hydrogen Economy Promotion and Hydrogen Safety Management Act in 2020. The Korean government has formulated and implemented various policies to build 6.2 million hydrogen fuel cell electric vehicles and 1200 hydrogen refueling stations by 2040 according to the national hydrogen economy roadmap, announced to revitalize the hydrogen economy [1].

A typical hydrogen refueling station consists of high-pressure compressors, storage tanks, a pressure ramp regulator, a pre-cooling system, and a hydrogen dispenser to dispense high-pressure hydrogen through a fueling nozzle that connects to a receptacle on the hydrogen fuel cell electric vehicle; an example of the layout of the hydrogen refueling

station is shown in Figure 1. Hydrogen refueling stations typically dispense high-pressure hydrogen gas of around 5 kg into the tanks of hydrogen vehicles, with wide span ranges of pressure and temperature [2,3]. A pre-cooling process to lower the temperature to −40 °C is done to prevent any substantial increase in the temperature of the compressed hydrogen gas during the fueling process, given that the pressure is as high as 70 MPa in the tank of the hydrogen vehicle. This process of fueling a hydrogen vehicle at a hydrogen fueling station is in accordance with the worldwide accepted fueling protocol SAE J2601 [4].

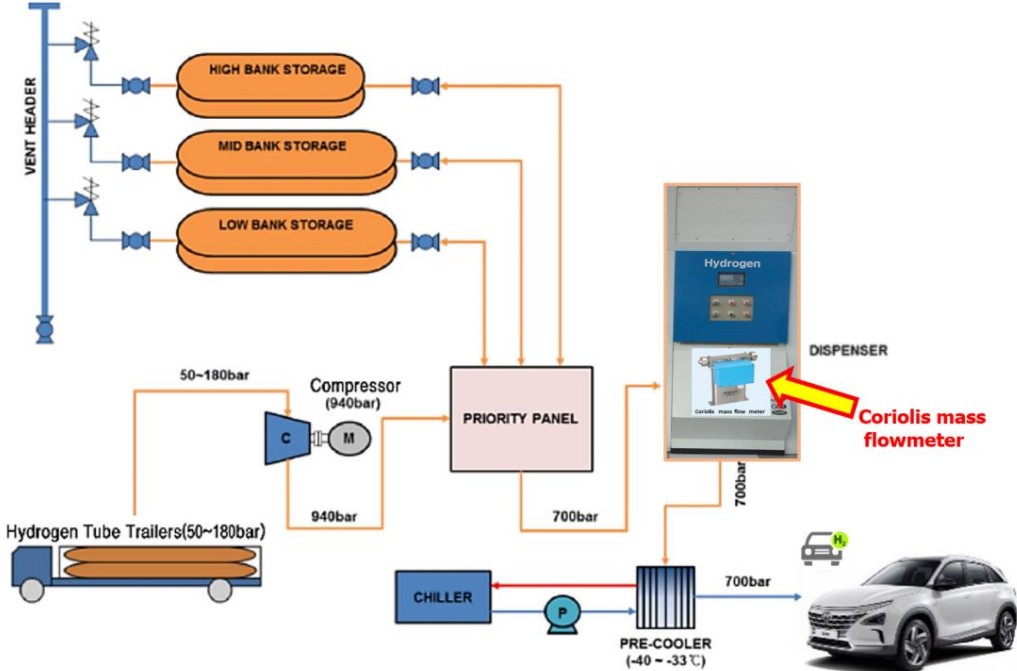

**Figure 1.** Sample layout for a hydrogen refueling station and the Coriolis mass flowmeter in the dispenser.

To measure the amount of hydrogen gas dispensed into the hydrogen vehicle for customer billing, a Coriolis mass flowmeter is typically installed inside the dispenser at a hydrogen refueling station. Because the hydrogen fueling process operates under a condition in which the flowrate, pressure, and temperature vary widely, a Coriolis mass flowmeter, which can directly measure the mass flow, is used to determine the hydrogen filling amount at a hydrogen refueling station. In order to ascertain the measurement reliability of hydrogen gas transferred to the hydrogen vehicle for customer billing, the international recommendation OIML R 139-1 requires certain accuracy levels for the flowmeters used at hydrogen fueling stations [5]. Table 1 shows the two classes of the maximum permissible error (MPE), which are defined as class 2 and class 4 to represent corresponding accuracy rate limits of 1.5% and 2% for these flowmeters. The MPE values are 2% to 5% for the two accuracy classes to ensure a complete measuring system during the type evaluation, verification, and in-service inspection processes that must take place when the stations are operating.

Several studies have investigated how to inspect and verify such accuracies of flow measurements under hydrogen-filling operating conditions at hydrogen refueling stations. It should also be noted that several portable gravimetric standards in national metrology institutes and the manufacturing companies of hydrogen dispensers in several countries have been developed for field verification according to the international recommendation for hydrogen refueling stations [6,7]. In a cooperative project for hydrogen vehicles by several European national metrology institutes, mobile gravimetric standards for verifying the measurements by the flowmeters inside the hydrogen dispensers were developed [8]. Several hydrogen refueling stations were tested in Europe using the gravimetric standard

developed during this project for the purpose of research related to the MPE acceptance criteria as defined in OIML R 139-1, which is the international recommendation [6–8].

**Table 1.** MPE value for the hydrogen meter and measuring system in OIML R 139-1 [5].

| Accuracy Class | MPE for the Meter (in % of the Measured Quantity Value) | MPE for the Completer Measuring System (in % of the Measured Quantity Value) | |
|---|---|---|---|
| Hydrogen Only | | Type Evaluation Initial or Subsequent Verification | In-Service Inspection Under Rated Operating Condition |
| 2 | 1.5 | 2 | 3 |
| 4 | 2 | 4 | 5 |

The Korea Research Institute of Standards and Science (KRISS) also developed a Hydrogen Field Test Standard (HFTS) that can be used for the field verification and calibration of the measurement accuracy of the mass flowmeters used at hydrogen refueling stations, as shown in Figure 2. The KRISS HFTS consists of three 52 L pressure cylinder tanks (type IV), a 300 kg weighing scale with a 0.5 g resolution, the Coriolis mass flowmeter, needle valves, temperature and pressure sensors, and a receptacle. The testing method is based on the gravimetric principle. The hydrogen delivered from the receptacle is passed through the Coriolis mass flowmeter, and the amount is determined by the weighing of the hydrogen gas collected in the pressure tank on the weighing scale.

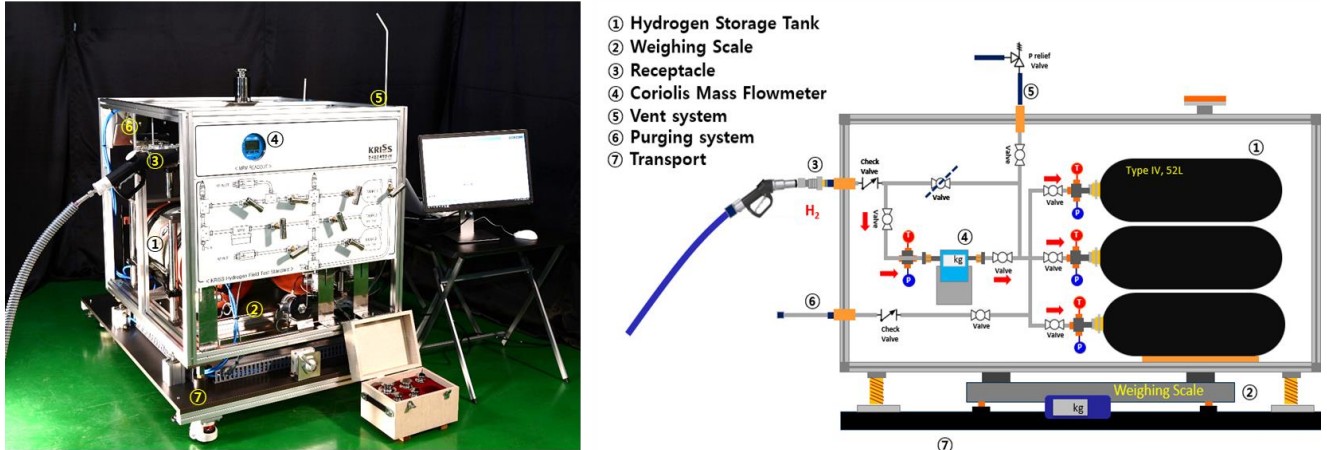

**Figure 2.** KRISS Hydrogen Field Test Standard (primary gravimetric system for flowmeters at hydrogen refueling stations).

However, only hydrogen vehicles are allowed access to hydrogen gas at hydrogen refueling stations in Korea under the Korean High-Pressure Gas Safety Control Act for all high-pressure gas, including hydrogen gas. A field test at a hydrogen refueling station using the KRISS HFTS could not be carried out due to legal and safety restrictions in Korea. Recently, regulations in the High-Pressure Gas Safety Control Act have been revised to enable filling processes for verification and research purposes at hydrogen refueling stations. Because administrative evaluations and certification are still required, actual field experiments with the KRISS HFTS have not been conducted thus far.

Currently, there is no traceable method by which to verify and calibrate the Coriolis mass flowmeters to be used at hydrogen refueling stations, except for a water calibration process as a conventional method for mass flowrate calibration. Typically, the Coriolis mass flowmeters in the hydrogen dispenser in Korea are calibrated with water by their manufacturers. In order to address these measurement challenges, several approaches to calibrate mass flowmeters for hydrogen refueling stations have been devised using alternative fluids

and matched operating conditions, such as the pressures and temperatures [9,10]. The performance and the measuring behavior of Coriolis mass flowmeters designed for use in hydrogen refueling stations were investigated with different fluids, including air and water with different pressure and temperature range conditions to simulate the fueling conditions of hydrogen gas when it is transferred to a hydrogen vehicle.

The aim of the present study is to develop the necessary methodologies and calibration facilities to allow a hydrogen refueling station to verify hydrogen flow metering to a suitable accuracy level under the challenging conditions of high pressure and a wide range of temperatures. In order to do this, the performance of Coriolis mass flowmeters under density conditions identical to those of hydrogen gas for refueling was assessed using compressed air as an alternative fluid at the high-pressure gas flow standard system of the Korea Research Institute of Standards and Science (KRISS). There is no facility or infrastructure to calibrate and verify Coriolis mass flowmeters with operating pressure conditions of up to 700 bar at hydrogen refueling stations. In the present study, we developed a high-pressure water flow test facility at KRISS to investigate the accuracy of Coriolis mass flowmeters with regard to operating pressure conditions using a substitute fluid such as water. The characteristics of the measurements of the Coriolis mass flowmeters to examine the pressure dependence of the supplied fluids were assessed in the pressure range of 2 bar to 700 bar.

## 2. Experimental Methods and Apparatus (Experimental Apparatus)

### 2.1. Density Matching Approach in High-Pressure Gas Flow Standard System of KRISS

The density-matching approach for the Coriolis mass flowmeters for use during hydrogen gas refueling was utilized with an alternative fluid in the high-pressure gas flow standard system of KRISS [11]. The conditions of the pressure and temperature of compressed air, 40 bar and 20 °C, were selected so that test could be carried out at a density of around 46 kg/m$^3$, equivalent to hydrogen gas at 700 bar and $-40$ °C. As shown in Figure 3, the high-pressure gas flow standard system at KRISS consists of two compressors, storage tanks with a capacity of 52.4 m$^3$, a temperature loop of 19.7 m$^3$, and control valves. This flow standard system is a blow-down type that generates a steady air flow in the pressure range of 1 to 5 MPa (10 to 50 bar). The maximum flowrate was 10,000 m$^3$/h at standard conditions (101.325 kPa and 293.15 K). The extended uncertainty of this standard system was 0.18% at the 95% confidence level. A sonic nozzle (ISO 9300) [12] was used as a reference meter; it was calibrated by a gravimetric standard with a fast-acting diverter and a weighing tank.

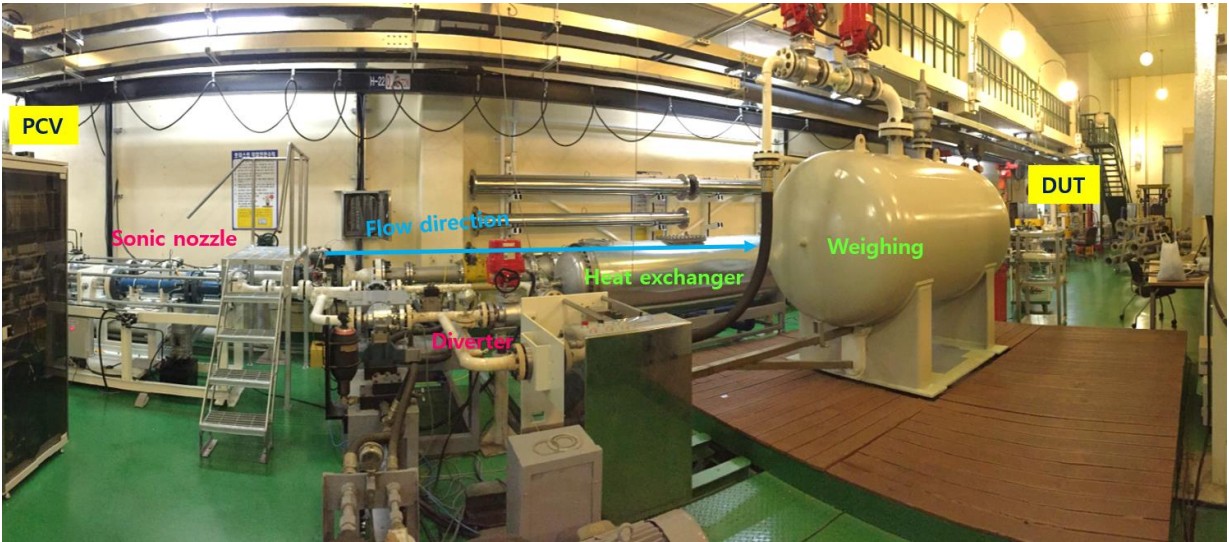

**Figure 3.** High-Pressure Gas Flow Standard System of KRISS.

As shown in Figure 4, The Coriolis mass flowmeter for the density-matching approach experiment was installed at the position of the test section between the second pressure control valves and the sonic nozzles. Sonic nozzles with throat diameters of 1.4 mm, 2.0 mm, 2.9 mm, and 3.6 mm were used to evaluate the accuracy of the measurement performance of the Coriolis mass flowmeter in the range of flowrates from 1.0 kg/min to 4.5 kg/min. The limits of maximum flowrate in the SAE J2601 protocol for light-duty vehicles is 3.6 kg/min [4]. Four sonic nozzles in the high-pressure gas flow standard system can cover up to the maximum flowrate specified by the SAE J2601. The inlet pressure condition of the Coriolis mass flowmeter was controlled by the first and second pressure control valves in the high-pressure gas flow standard system of KRISS at 40 bar, which is a density condition identical to that during hydrogen gas refueling into vehicles at a hydrogen station at 700 bar. The operating temperature condition for the ambient temperature (around 20 °C) in the density-matching approach was controlled and stabilized by the temperature control loop with a capacity of 19.7 m$^3$, located upstream of the Coriolis mass flowmeter and sonic nozzles.

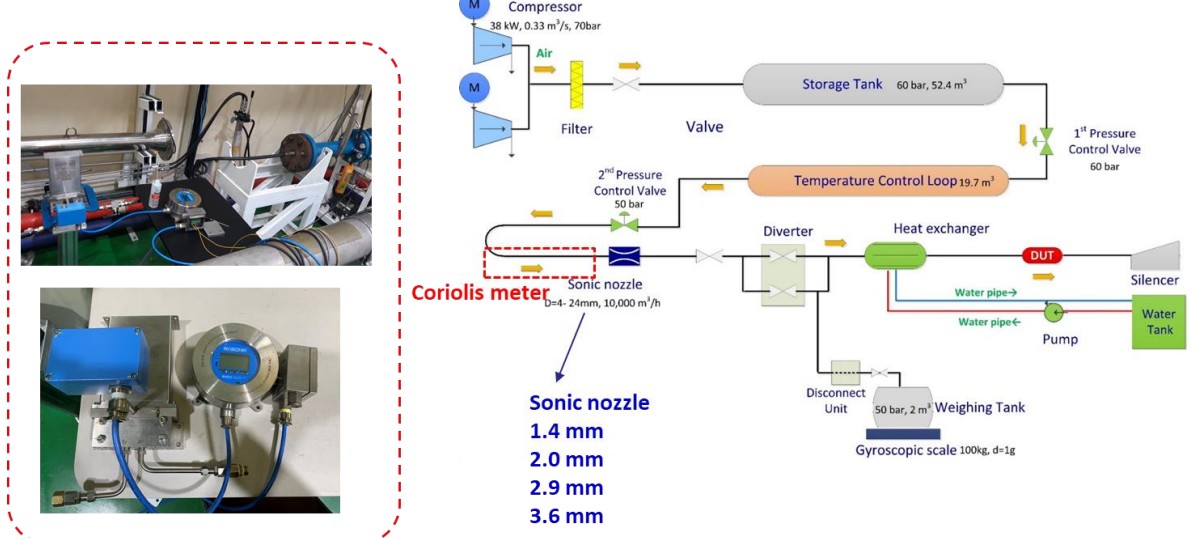

**Figure 4.** Installation of the Coriolis mass flowmeter for the density-matching experiments.

## 2.2. Pressure Matching Approach in High-Pressure Water Flow Test Facility of KRISS

To investigate the pressure dependence of the mass flowmeter of a hydrogen refueling station, a high-pressure water flow test facility was developed, as shown in Figure 5. This facility consists of a high-pressure liquid pump (Ecotech-MR300) (up to 1000 bar), two buffer tanks (80 L) to reduce the transmission of excessive vibration and pulsations from the pump, and two pressure regulators (PRV-1000S, DASUNG ACE Co., Ltd., Seoul, Republic of Korea). The first pressure regulator controls the water pressure in the pipeline from 700 bar to 200 bar, while the second pressure regulator controls the pressure in a range of 200 bar to 2 bar. One Coriolis mass flowmeter was installed upstream of the two pressure regulators as the device under test (DUT) to evaluate the measurement characteristics with regard to the operating pressure conditions. The other Coriolis mass flowmeter was installed downstream of the two pressure regulators as a reference meter traceable to the KRISS National Water Flow Standard System, which is located at the end of the line of the high-pressure water flow test facility. The KRISS Water Flow Standard System is a gravimetric system operated by flow diverters that utilizes the flying-starting-and-finish method. The weighbridges can collect water into the weighing tank, which has a weighing capacity of 60 kg with a resolution of 0.1 g (Mettler Toledo, Columbus, OH, USA). To ensure the pressure effect of the Coriolis mass flowmeter installed upstream of the pressure regulators, the inlet water pressure was controlled by a high-pressure liquid pump and

two regulators. The flowrate condition for testing from 1 to 5 kg/min was controlled by a control valve downstream of the high-pressure water flow facility.

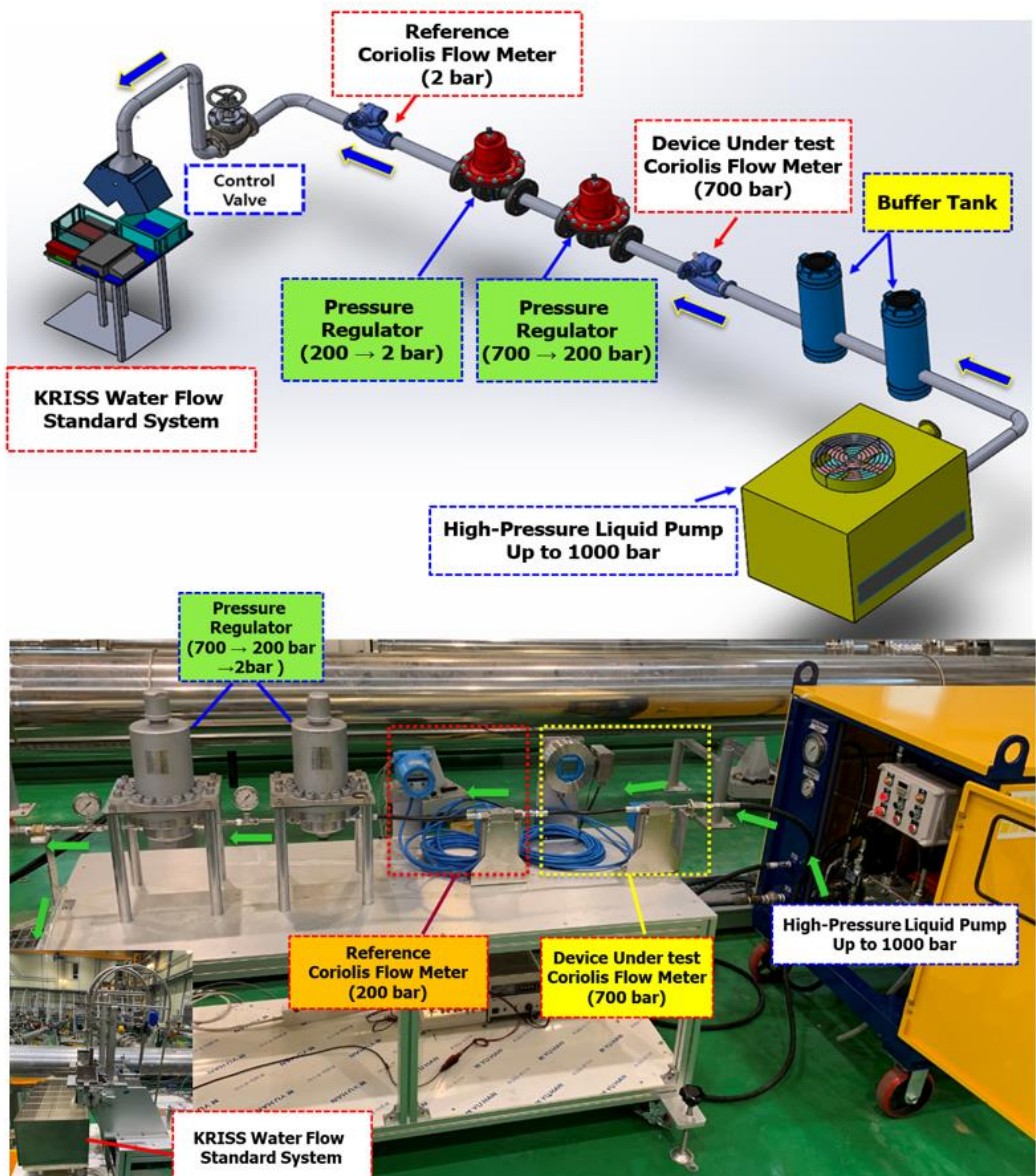

**Figure 5.** High-Pressure Water Flow Test Facility of KRISS.

## 3. Results and Discussion

### 3.1. Test Results for Density Matching Approach

The Coriolis mass flowmeter for the test selected here is the type most frequently used in hydrogen refueling stations in Korea. The flowmeter is designed for flowrates from 0.2 kg/min to 10 kg/min. The maximum pressure is up to 1050 bar, and the temperature ranges are from −50 °C to 120 °C. The autoclave 3/8″ MP process connections were used for the installation. In order to ensure reliable zero-flow stability, zero-point adjustments of the Coriolis mass flowmeter were carried out under the operating pressure and temperature conditions in the present study. The measurement characteristics of the Coriolis mass flowmeter for the density-matching approach were assessed using reference sonic nozzles with different throat diameters and working pressure conditions in the High-Pressure Gas Flow Standard System of KRISS. The sonic nozzles, for which the diameters of the throat are 1.4 mm, 2.0 mm, 2.9 mm, and 3.6 mm, used for the reference flowmeter at different working pressures were calibrated with the gravimetric method using a diverter, a weighing tank,

and a gyroscopic scale, all of which were traceable to the national SI-unit standard of KRISS, as shown in Figure 4. The four sonic nozzles with different throat diameters as used here are capable of covering a flowrate range of 1.0 kg/min to 4.5 kg/min, depending on the working pressure condition. The measurement performance tests of the Coriolis mass flowmeter at the four different working pressure conditions of 10, 20, 30, and 40 bar were conducted with the four sonic nozzles of the High-Pressure Gas Flow Standard System of KRISS.

Figure 6 shows the test results of the Coriolis mass flowmeter at the different working pressures from 10 to 40 bar when using the four sonic nozzles for the flowrate range from 1.0 kg/min to 4.5 kg/min. The pressure conditions of the incoming air from 10 to 40 bar correspond to the different density conditions of 12 kg/min$^3$ to 46 kg/min$^3$. At each measuring point, three repeated measurements were carried out in different pressure and flowrate conditions. The error (%) is defined as the relative difference between the measurement results of the Coriolis mass flowmeter and the reference sonic nozzles at each pressure and flowrate condition. The error of the Coriolis mass flowmeter ranged from $-2.0\%$ to $0.5\%$ within the flowrate range of 1.0 kg/min to 4.5 kg/min. Large errors occurred at a low flowrate, below 1.0 kg/min, from 10 to 30 bar, which is typical behavior of Coriolis mass flowmeters. Above 1.0 kg/min, the error was approximately linear, and the average error was within $\pm0.2\%$. This tendency toward error of the Coriolis mass flowmeter with different pressure conditions is similar to previous research [9].

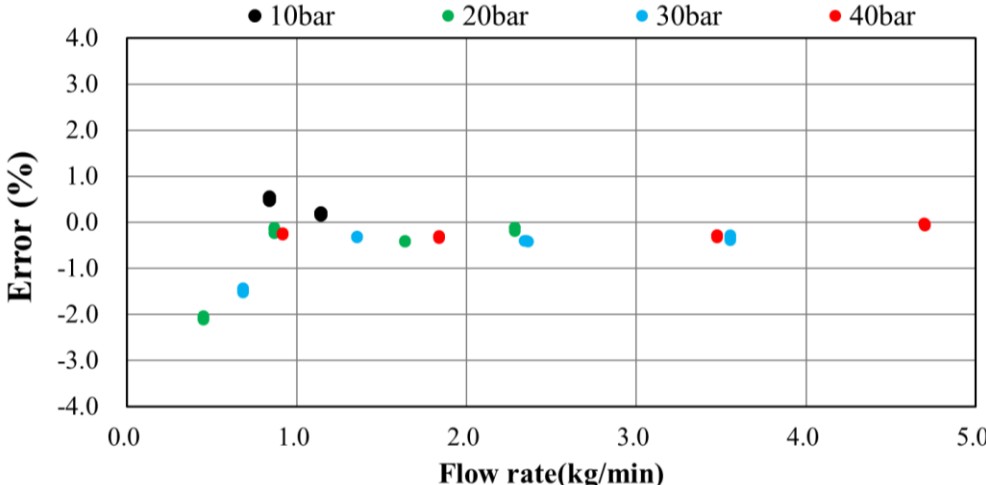

**Figure 6.** Measurement results of the Coriolis mass flowmeter with different pressure and flowrate conditions.

Figure 7 shows the test results of the Coriolis mass flowmeter at 40 bar and at an ambient temperature (around 20 °C) with the density set to approximately 46 kg/m$^3$, equivalent to hydrogen gas at 700 bar and $-40$ °C. The error of the Coriolis mass flowmeter was within $\pm0.30\%$ at flowrates from 1 kg/min to 4.5 kg/min, and the average error was within $\pm0.20\%$. The repeatability of three measurements at each flowrate was less than $\pm0.02$. These results are similar to the typical water calibration results of Coriolis mass flowmeters installed at hydrogen dispensers at stations. The results are from the density-matching approach of the Coriolis mass flowmeters for hydrogen gas refueling conducted with an alternative fluid. However, in the present study the operating temperature for the density-matching approach was limited to an ambient temperature (around 20 °C). Previous research [8,9] shows the significant temperature dependency of the performance of the Coriolis mass flowmeters in the wide temperature range of $-40$ °C to 20 °C. In Korea, the Coriolis mass flowmeters in hydrogen refueling stations are installed near the upstream of the pre-cooler, which means that the operating conditions of the flowmeter are not as cold as the conditions after the pre-cooler. Nevertheless, the temperature dependency of the Coriolis mass flowmeters should be addressed in the density-matching approach. If

temperature dependency of the operating condition of the Coriolis mass flowmeters is verified later, the density-matching approach can represent a potential method by which to perform calibration tests of Coriolis mass flowmeters at hydrogen refueling stations that do not have calibration facilities for hydrogen gas at 700 bar and −40 °C.

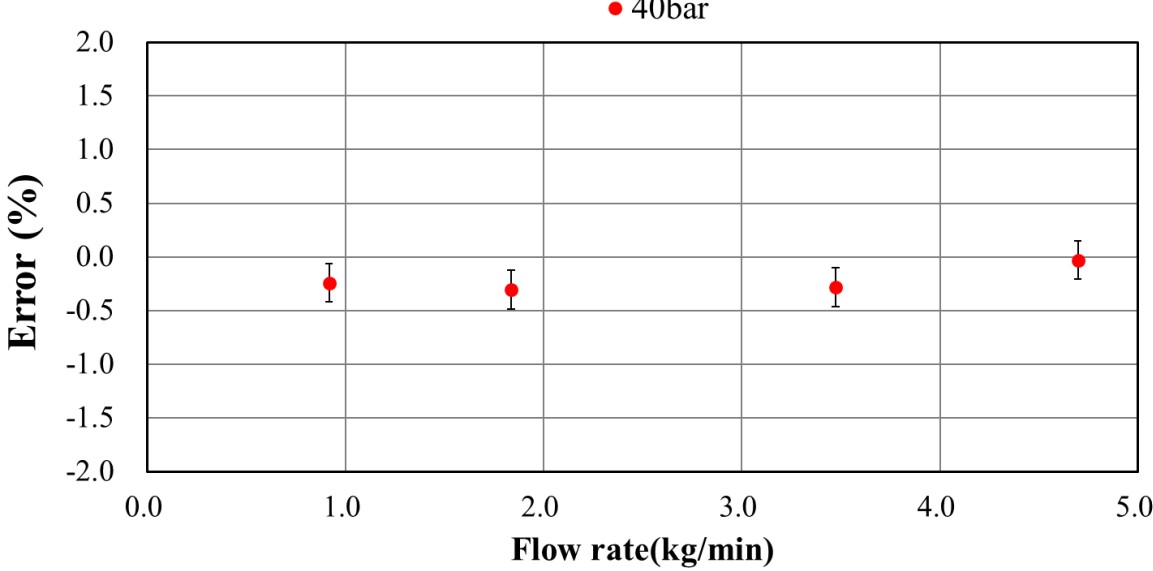

**Figure 7.** Measurement results of Coriolis mass flowmeters with the density-matching approach.

### 3.2. Test Results for Pressure-Matching Approach

The measurement characteristics of the Coriolis mass flowmeter for the pressure-matching approach were assessed using the high-pressure water flow test facility to investigate the pressure dependence of the flowmeters at hydrogen refueling stations, as shown in Figure 5. The measurement test of the pressure-matching approach was conducted with the same Coriolis mass flowmeter used in the test with the density-matching approach, shown in Figure 4. The Coriolis mass flowmeter was installed in a position where the inlet pressure of the supplied water could be controlled by a high-pressure liquid pump and two pressure regulators. The pressure range of the supplied water to the installed Coriolis mass flowmeter, adjustable in the high-pressure water flow test facility, is 2 bar to 700 bar, which is equivalent to the operating pressure conditions at a hydrogen refueling station. The flowrate conditions of the measurement test at the installed Coriolis mass flowmeter were 1 kg/min to 5 kg/min, as controlled by a control valve downstream of the high-pressure water flow facility.

Figure 8 presents the test results of the Coriolis mass flowmeter when the inlet pressure of the supplied water is set to 2 bar. Five different mass flowrates ranging from 1.0 kg/min to 5.0 kg/min were used, as this is the most relevant mass flowrate range regarding the hydrogen refueling process. At each measuring point, three repeated measurements were taken. The error (%) is defined as the relative difference between the measurement results of the Coriolis mass flowmeter installed upstream for the pressurized condition and for the other Coriolis mass flowmeter installed downstream of the two pressure regulators. The Coriolis mass flowmeter installed at the downstream location was used as the reference meter, given its traceability to the KRISS National Water Flow Standard System. The error of the Coriolis mass flowmeter was within ±0.10% at flowrates from 1.0 kg/min to 5.0 kg/min, and the average error was ±0.10% The repeatability of three measurements at each flowrate was less than ±0.04%.

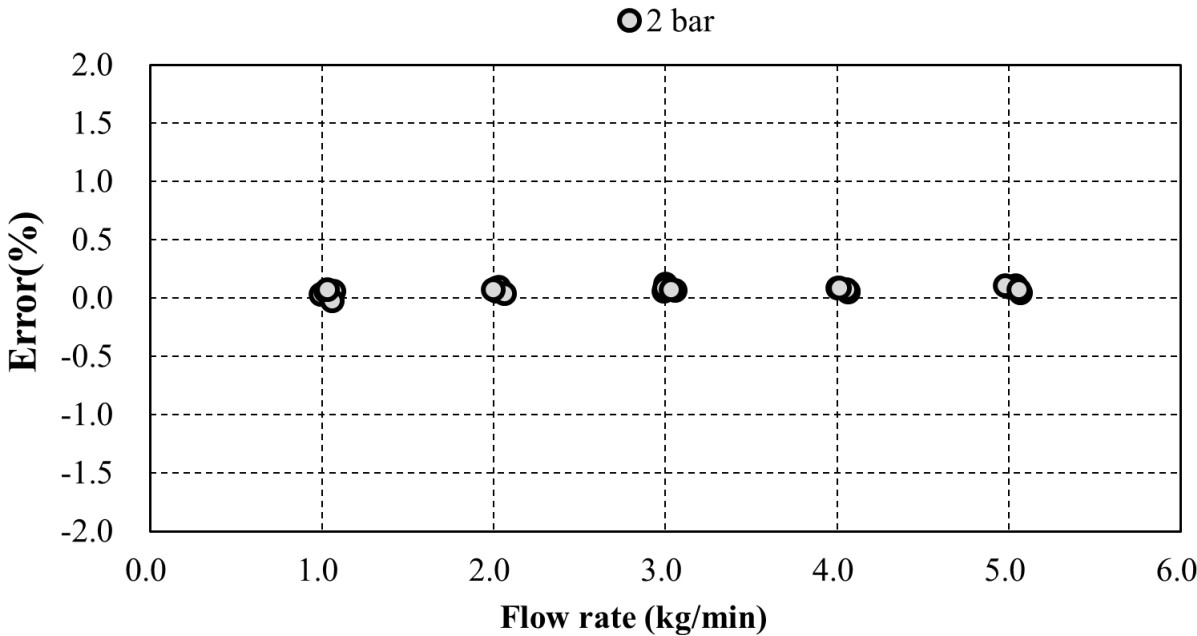

**Figure 8.** Measurement results of the Coriolis mass flowmeter at the inlet pressure condition of 2 bar for the pressure-matching approach.

Figure 9 shows the test results of the Coriolis mass flowmeter when the inlet pressure of the water supplied to the flowmeter was set to 100 bar. The condition of the inlet pressure was increased and controlled by a high-pressure liquid pump and pressure regulators at flowrates of 1.0 kg/min to 5.0 kg/min. In the comparison between the measurement results at 2 bar and 100 bar for the inlet pressure of the water supplied to the Coriolis mass flowmeter, no pressure dependence could be identified. It is interesting to note that when the inlet pressure to the Coriolis mass flowmeter was increased up to 100 bar, the calibration results of the flowmeter were similar to calibration results found by the manufacturer at ambient pressures with water.

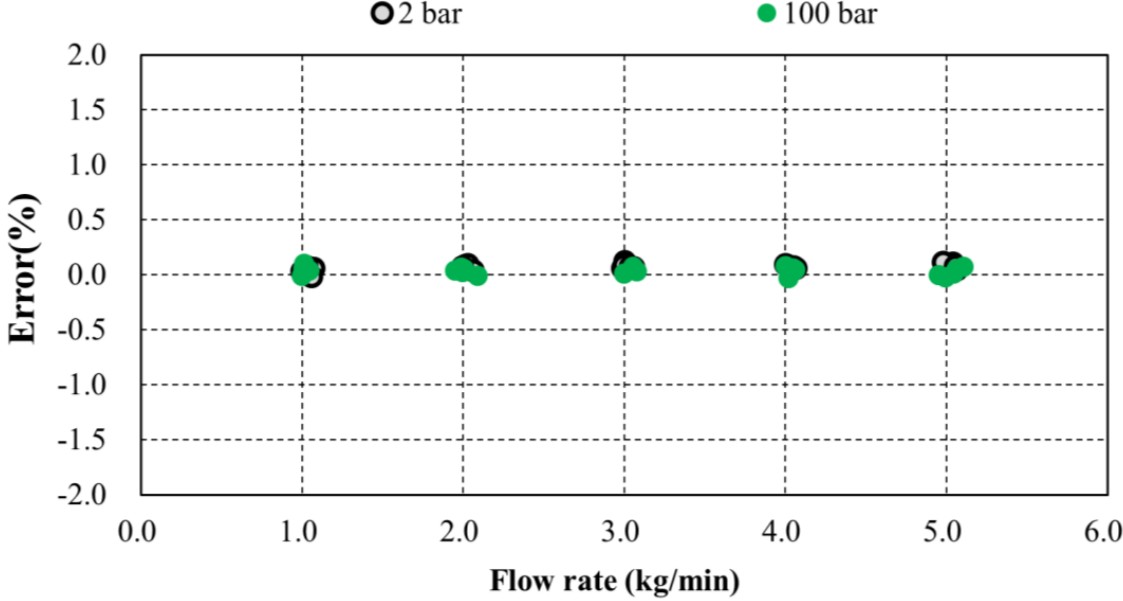

**Figure 9.** Measurement results of the Coriolis mass flowmeter at the inlet pressure condition of 2 bar and 100 bar.

Figure 10 shows the measurement results of the Coriolis mass flowmeter when the inlet pressure of the water supplied to the flowmeter was 700 bar, equivalent to the operating pressure conditions at hydrogen refueling stations. The measurement characteristics of the Coriolis mass flowmeter for the pressure-matching approach were investigated in a pressure range of 2 bar to 700 bar at flowrates in the range of 1.0 kg/min to 5.0 kg/min. The error of the Coriolis mass flowmeter was within ±0.10% at flowrates from 1.0 kg/min to 5.0 kg/min, and the average error was ±0.10%. The measurement results of the Coriolis mass flowmeter for the pressure-matching approach show no detectable dependence of the pressure on the accuracy of measurement performance in the pressure range of 2 bar to 700 bar. It can be assumed that the pressure corrections implemented by the manufacturer of the Coriolis mass flowmeter work well in the pressure and flowrate ranges utilized in this study.

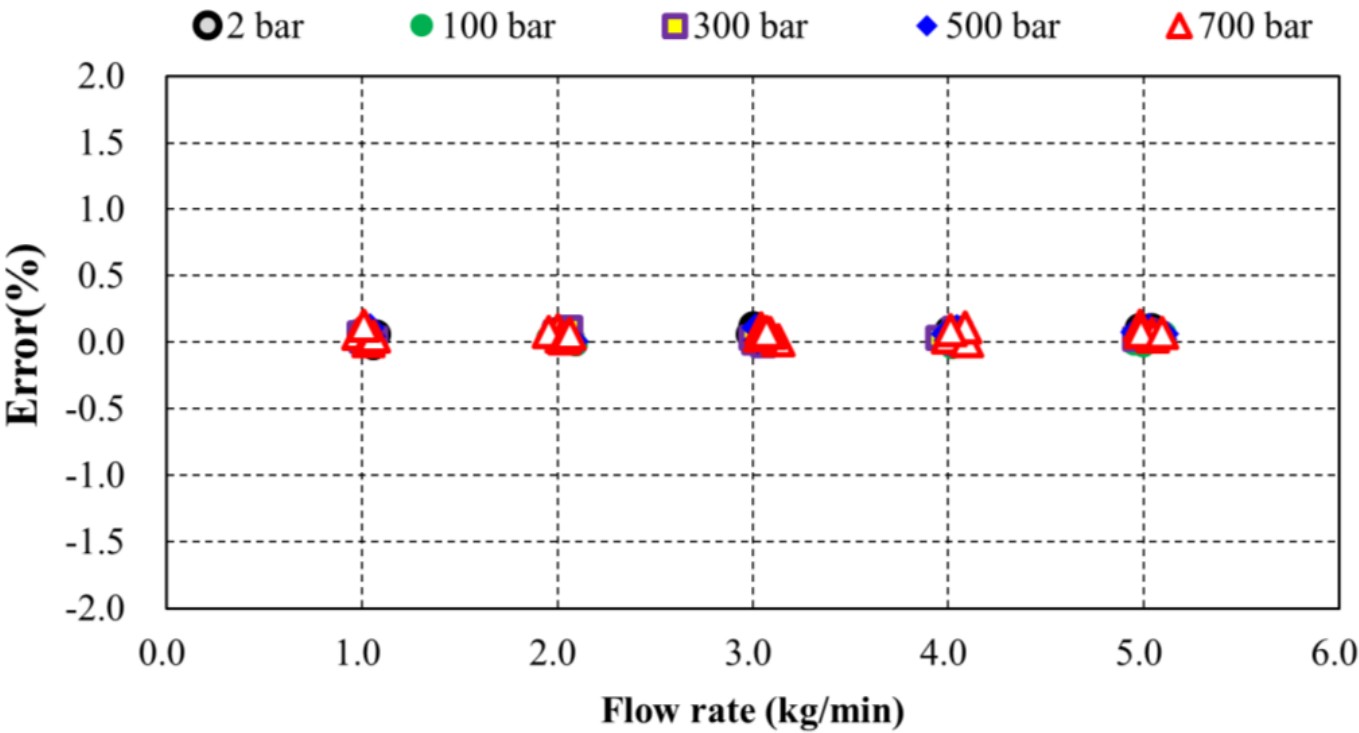

**Figure 10.** Measurement results of the Coriolis mass flowmeter at the inlet pressure condition from 2 bar to 700 bar.

*3.3. Matching Approaches and Hydrogen Field Test Standard*

Figure 11 shows the comparison of the calibration test results of the Coriolis mass flowmeters by the density-matching approach and the pressure-matching approach under the vehicle hydrogen refueling condition at hydrogen refueling stations. The two measurements were taken with air at a pressure of 40 bar with the high-pressure Gas Flow Standard System of KRISS and with water at a pressure of 2 bar to 700 bar using the high-pressure water flow test facility. As shown in Figure 11, the test results between the density-matching approach and the pressure-matching approach were within the corresponding measurement uncertainty rates. The difference in the errors between the results of the two approaches is due to the speed of sound being different in air and water. Although the medium was not hydrogen gas but was instead a substitute fluid, i.e., compressed air or water, and the pressure conditions differed from those when refueling at a hydrogen station, this is the first study in Korea to investigate the measurement performance capabilities of the Coriolis flowmeters for use at hydrogen stations with alternative calibration by a density-matching approach and a pressure-matching approach. In Korea, the Coriolis mass flowmeters in the hydrogen refueling station are installed near the upstream of the

pre-cooler; this means that the operating condition of the flowmeter is not as cold as the condition after the pre-cooler (−40 °C). Nevertheless, the temperature dependency on the Coriolis mass flowmeter should be addressed in the density and pressure-matching approaches. If temperature dependency for the operating condition of the Coriolis mass flowmeters is addressed in the future, with the development of new facilities for temperature control in the operating conditions for flowmeters, the results of these approaches can be presented as the best means of accurately calibrating Coriolis mass flowmeters for hydrogen refueling stations where there is no facility to calibrate them with hydrogen gas at 700 bar and −40 °C.

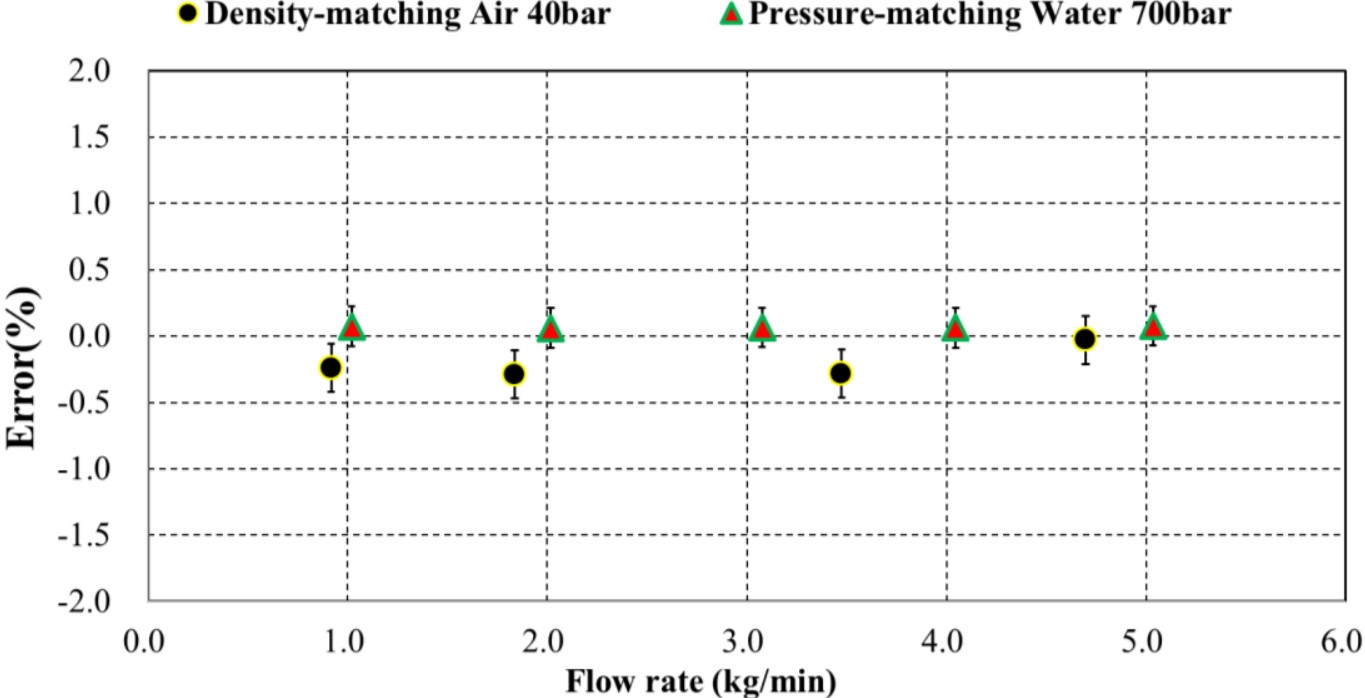

**Figure 11.** Comparison between the test results of Coriolis mass flowmeters for the density-matching approach and the pressure-matching approach.

According to the newly enforced regulation of the High-Pressure Gas Safety Control Act, KRISS is developing a new Hydrogen Field Test Standard, as shown in Figure 12. This new facility utilizes a trailer and is designed to maintain safe and strict explosion-proof rules. Field experiments with KRISS HFTS are planned at hydrogen refueling stations for the first time in Korea in the near future. The Coriolis mass flowmeter used in the density-matching approach and pressure-matching approach will be installed at the KRISS HFTS during field experiments at the hydrogen refueling stations. This will allow a comparison between the test results from hydrogen gas and alternative fluids such as compressed air or water with the density- and pressure-matching approaches.

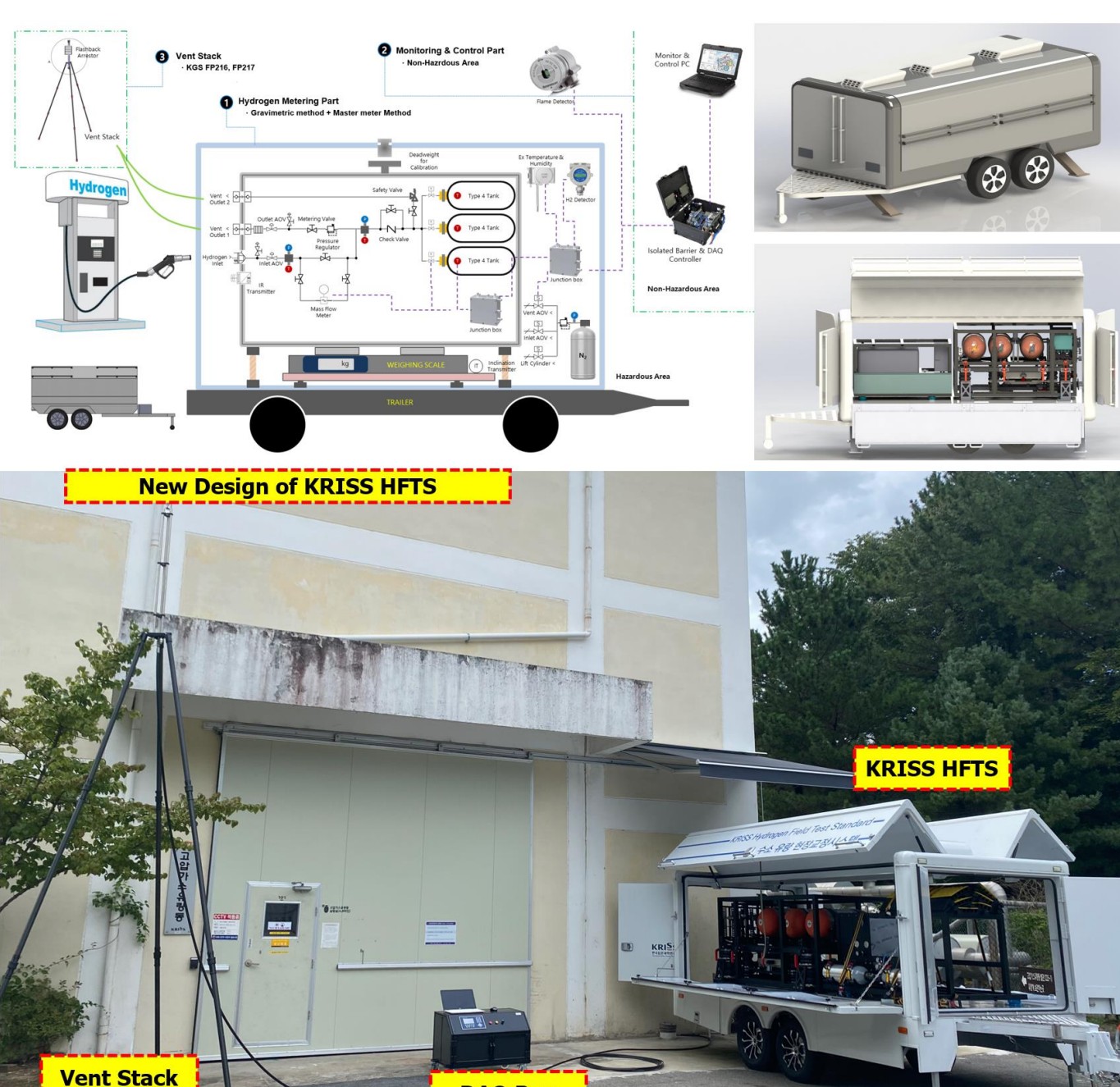

**Figure 12.** New development of the KRISS Hydrogen Field Test Standard according to the Korean Gas Safety Act.

## 4. Conclusions

Hydrogen fuel cell electric vehicles are emerging as a means of transportation using renewable and carbon-free energy, especially considering global warming and air pollution problems in many places of the world. In relation to the supply and distribution of hydrogen vehicles, there is increasing national support for the development of the required hydrogen infrastructure, specifically hydrogen refueling stations, in many countries. Hydrogen fuel cell electric vehicles are typically refueled at a wide range of temperatures (−40 °C to 85 °C) and at a high pressure (up to 875 bar) at hydrogen stations in accordance with globally accepted standards. Currently, there is no traceable method by which to verify and calibrate

the Coriolis mass flowmeters to be used at hydrogen refueling stations, except for a water calibration process as a conventional method for mass flowrate calibrations.

In the present study, we developed the necessary methodologies and calibration facilities to allow a hydrogen refueling station to verify hydrogen flow metering to a suitable accuracy level under the challenging conditions of high pressure and a wide range of temperatures. In order to do this, the performance of Coriolis mass flowmeters under density conditions identical to those of hydrogen gas for refueling was assessed using compressed air as an alternative fluid in the high-pressure gas flow standard system of KRISS. The results of tests of a Coriolis mass flowmeter at 40 bar and at an ambient temperature (around 20 °C) with the density set to approximately 46 kg/m$^3$, equivalent to hydrogen gas at 700 bar and −40 °C, showed that the average errors were within ±0.2%, from 1 kg/min to 4.5 kg/min.

We also developed a high-pressure water flow test facility at KRISS to investigate the accuracy of Coriolis mass flowmeters with regard to typical operating pressure conditions using a substitute fluid, in this case, water. The measurement characteristics of the Coriolis mass flowmeter during this pressure-matching approach were investigated, showing an error of less than −0.2% in a pressure range of 2 bar to 700 bar at flowrates in the range of 1.0 kg/min to 5.0 kg/min, with no detectable dependence of the pressure on the accuracy of the measurement. It can be assumed that the pressure corrections implemented by the manufacturer of the Coriolis mass flowmeter work well in the pressure and flowrate ranges utilized in this study. The calibration test results between the density-matching approach and the pressure-matching approach were within the corresponding measurement uncertainty rates. The difference in the errors between the results of the two approaches is due to the speed of sound being different in the air and water. As the first study in Korea, a comparison of the calibration test results of the Coriolis mass flowmeters by the density-matching approach and the pressure-matching approach under vehicle hydrogen refueling conditions shows that these approaches can be presented as a potential means of calibrating Coriolis mass flowmeters for hydrogen refueling stations in the absence of a facility to calibrate them with hydrogen gas, at 700 bar and −40 °C, if temperature dependency for the operating condition of the Coriolis mass flowmeters were further verified. In addition, to apply these approaches, the field performance test for the Coriolis mass flowmeter tested in the calibration laboratory should be conducted at the hydrogen refueling station first. The type evaluation and subsequent verification with these density and pressure-matching approaches should be performed after more parameters, including the temperature dependency of operating conditions, were clearly addressed.

Finally, KRISS developed a Hydrogen Field Test Standard (HFTS) that can be used for field verifications and calibrations of the measurement accuracy of the mass flowmeters used at hydrogen refueling stations. According to the newly enforced regulation of the High-Pressure Gas Safety Control Act, KRISS is developing a new facility in the form of a trailer that will maintain safe and strict explosion safety rules. Field experiments with the KRISS HFTS are planned at hydrogen refueling stations for the first time in Korea in the near future. The Coriolis mass flowmeter tested in the high-pressure gas flow standard system and high-pressure water flow facility will be installed and tested by KRISS HFTS at the hydrogen refueling station.

**Author Contributions:** Methodology and writing, W.K.; experiments, J.S. and B.Y.; data analysis, S.K., S.Y. and W.H.; writing—review and editing, U.B. All authors have read and agreed to the published version of the manuscript.

**Funding:** This research was funded by Korea Institute of Energy Technology Evaluation and Planning (KETEP), granted financial resources from the Ministry of Trade, Industry & Energy, Republic of Korea (No. 20202910100060).

**Acknowledgments:** This study was carried out as a project of the 'Development and Empirical study of Legal Metrology Standard Model for Preventing the Explosion Accident by Over-charge in a Hydrogen Refueling Station' supported by the Korea Institute of Energy Technology Evaluation

**Conflicts of Interest:** The authors declare no conflict of interest.

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
