# Peer review of "Investigation of a Calibration Method of Coriolis Mass Flowmeters by Density- and Pressure-Matching Approaches for Hydrogen Refueling Stations"

_applsci, doi:10.3390/app122412609_

Round 1

Reviewer 1 Report

Please see the attached file for comments and improvements.

Author Response

Responses to Reviewer 1

“Investigation of a Calibration Method of Coriolis Mass Flowmeters by Density and Pressure-Matching Approaches for Hydrogen Refueling Stations” by Woong Kang, Jinwoo Shin, Byungro Yoon, Sunghee Kil, Sangsik Yim, Wonguk Han and Unbong Baek

We are really thankful to the reviewer for his productive comments. We are pleased that the paper had generally received favorable and positive evaluations from the reviewer.

We have prepared a revised manuscript in accordance with the suggestions of the reviewer. As to the specific responses in the revised paper, we would like to note the following modifications. (blue letters)

Reviewer #1’s comment:

The authors present a calibration method for flowmeters. The special challenges at fuel stations are a wide range of operating parameters and the requirement of using different fluids than hydrogen. The article describes the growing hydrogen mobility market in the introduction with a focus on Korea. It is stated the many primary standards have been developed in Europe to perform field verifications of hydrogen refueling stations and a KRISS HFTS have been designed and built but could not be used due to regulation (at the first stage). This is most likely the reason of looking for an alternative procedure with inert gases (faster, safer, cheaper).

Two experimental setups are described and used for this paper. A gas facility (0.18% k=2) and a water facility (where uncertainties are not provided). The gas facility has sonic nozzles as transfer standards. The pressure was set at 40 bar and 20°C to get the same density during a filling at 700Bar -40°C of hydrogen. The pressure dependency is also verified with a water calibration facility at several pressures. The results show typical Coriolis flow meter error curve (both with gas and water) and are rather small and within the OIMLR139 MPE for initial verification for the stable temperature and pressure conditions. The authors provide conclusions and give information about the new trailer for HFTS to fulfil the ATEX regulation.

Comments:

The topic of this paper is important since the hydrogen market is growing and initial and subsequent verifications in the world are needed in the upcoming months. The approach of the pressure / density dependency is correct at the first stage, but it is not enough. Indeed, during the refueling the pressure / temperature change rapidly at the beginning of the process (from ambient to -35°C in 30/45 sec over 4 min of refueling). This is where the mass flow meter can have significant deviations. The temperature dependency must be verified before validating this alternative method to calibrate a hydrogen mass flow meter. Metrohyve (I) project did the same density / pressure equivalence (papers have been cited in this paper) but they also checked the temperature dependency (at METAS), and they showed that the accuracy can significantly be modified (+/-5%)

I suggest the authors to perform the temperature dependency tests (-40° / -20°C / 0°C / 20°C –changing temperature if possible (with another transfer standard than CFVN)) before giving any recommendations on these alternative procedures. It will greatly depend on the mass flow meters manufacturer and the temperature compensation function implemented.

  • As pointed out by the reviewer, in the present study the operating temperature for the density matching approach was limited to an ambient temperature (20 ℃). I totally agree with the comments of the reviewer about the importance of temperature dependency of the performance for the Coriolis mass flowmeter as shown in the previous research [8,9]. Even though in Korea, the Coriolis mass flowmeters in the hydrogen refueling station are almost installed at the upstream of the pre-cooler, which means that the operating condition of the flowmeter is not as cold as the condition after the pre-cooler (-40 ℃). Nevertheless, the temperature dependency on the Coriolis mass flowmeter should be addressed in the density matching approaches for the hydrogen refueling stations

  • This is the first step in Korea to investigate the measurement performance capabilities of the Coriolis mass flowmeter with alternative calibration method. We focused to develop the two different matching approaches and to check the consistency the test results from each high-pressure gas and water flow system. As pointed out by the reviewer, for the direct next step, we will test the temperature dependency for flowmeter in the high-pressure gas flow standard system a by developing a new facility for the temperature control of working range for hydrogen refueling station as soon as possible in the near future. In addition, if possible, we will try to do international comparison study with previous research group for the temperature dependency of performance for the Coriolis mass flowmeter in the hydrogen refueling stations.

I fear that readers could consider performing subsequent verification with air / nitrogen at 40 bar / 20°C where all the evidence of equivalence are not clearly demonstrated.

  • To avoid the misunderstanding of the results from the density matching approach with limited temperature condition for readers who consider performing subsequent verification, we have clearly revised the manuscript with the detailed explanation in the chapter 1 Test Results for Density Matching Approach (page 8), 3.3 Matching Approaches and Hydrogen Field Test Standard (page 11) and 4 Conclusion (page 13) with blue letters.

* Following the comments of reviewer, we carefully checked the spell in the English language.

We feel that the comments of the reviewer were stimulating and productive. These led to improvements in the revised manuscript. We are appreciative of the reviewer’s efforts for our paper.

Reviewer 2 Report

The paper presents experimental data based on measurements of Coriolis meters using various fluids (air and water) at different pressures. Identical measurements have been performed in the past and published, references are quoted in this paper.

This paper presents new data but no new findings and does not compare the new data to previously published data. Unless the authors perform a proper analysis by comparing their data with previously published data to confirm the findings of the other publications, this paper is not suited for publication. 

Author Response

Responses to Reviewer 2

“Investigation of a Calibration Method of Coriolis Mass Flowmeters by Density and Pressure-Matching Approaches for Hydrogen Refueling Stations” by Woong Kang, Jinwoo Shin, Byungro Yoon, Sunghee Kil, Sangsik Yim, Wonguk Han and Unbong Baek

We are really thankful to the reviewer for his productive comments. We are pleased that the paper had generally received favorable and positive evaluations from the reviewer.

We have prepared a revised manuscript in accordance with the suggestions of the reviewer. As to the specific responses in the revised paper, we would like to note the following modifications. (blue letters)

Reviewer #2’s comment:

The paper presents experimental data based on measurements of Coriolis meters using various fluids (air and water) at different pressures. Identical measurements have been performed in the past and published, references are quoted in this paper.

This paper presents new data but no new findings and does not compare the new data to previously published data. Unless the authors perform a proper analysis by comparing their data with previously published data to confirm the findings of the other publications, this paper is not suited for publication.

  • As pointed out by the reviewer, in the present study there are no direct comparison with previously published data. But, for this study we participated the cooperative project for hydrogen vehicles of European national metrology institutes [8] as an international partner from 2017 to 2020. As the cooperation of these project, our test results from the density-matching approach in the high-pressure gas flow system were used in their final report and conference paper as below figures.

* Good Practice Guide: Calibration and validation flow meters used at HRSs for quantifying hydrogen dispensed into vehicles (EMPIR Project Report)

        - page 30, 31(figure 19)

* FLOMEKO 2019 Proceeding: Air and Nitrogen Testing Coriolis Flow Meters Designed for Hydrogen Refueling Stations

 - page 1, 7 (figure 12)

  •  The test results used in these reports and conference paper were conducted with different Coriolis mass flowmeters used in the present study. Therefore, we cannot directly compare our data with the other research data. But, the tendency of test result by the density matching approach are very similar to the results of previously published data in the same pressure and temperature condition. We have revised the manuscript with the explanation about the comparison with published paper [9] with blue letters in the chapter 3.1 Test Results for Density Matching Approach (page 7). 
  • For the pressure-matching approach, there is previously research data [10] which published by the member of cooperative project for hydrogen vehicles. We didn’t the comparison the performance of the Coriolis mass flowmeter for pressure-matching approach with their system. But, we continue to discuss with international comparison for this research between two systems. In their published paper [10], the future plan for this bi-lateral international comparison were described in detail as below figure.

  • This is the first step in Korea to investigate the measurement performance capabilities of the Coriolis mass flowmeter with an alternative calibration method. We focused to develop the two different matching approaches and to check the consistency the test results from each high-pressure gas and water flow system. As pointed out by the reviewer, for the direct next step, we will conduct the comparison study with another research group by sending the Coriolis mass flowmeter. Because it is necessary to directly compare the test results with other previous data, same Coriolis mass flowmeter should be used. That’s why there is no direct comparison with previously published data. So, we sincerely ask you to reconsider your opinion on the publication of our paper. 

* Following the comments of reviewer, we carefully corrected and revised English language in this manuscript.

We feel that the comments of the reviewer were stimulating and productive. These led to improvements in the revised manuscript. We are appreciative of the reviewer’s efforts for our paper.

We sincerely ask you to reconsider your opinion on the publication of our paper with these response and revised manuscripts.

Reviewer 3 Report

Title: Investigation of a Calibration Method of Coriolis Mass Flowmeters by Density and Pressure-Matching Approaches for Hydrogen Refueling Stations

The authors present a calibration method for flowmeters. The special challenges at fuel stations are a wide range of operating parameters and the requirement of using different fluids than hydrogen.

Recommendation: minor changes

Minor concerns

  1. Fig. 8: Was the standard deviation in calculated as uncorrected or corrected standard deviation (divided by N or divided by (N-1))?

  2. Page 8, bottom paragraph: What made the decision to use three measurements for the repeatability test?

  3. Based on the results in Figs. 7-12: what is the classification of the flowmeter according to Table 1?

  4. Would a higher number of measurements for the repeatability test change the classification?

  5. Reference [5] contains OIML R139-1 in its appendix, which shows a list of requirements, additional to Table 1. Are these requirements all fulfilled?

Author Response

Responses to Reviewer 3

“Investigation of a Calibration Method of Coriolis Mass Flowmeters by Density and Pressure-Matching Approaches for Hydrogen Refueling Stations” by Woong Kang, Jinwoo Shin, Byungro Yoon, Sunghee Kil, Sangsik Yim, Wonguk Han and Unbong Baek

We are really thankful to the reviewer for his productive comments. We are pleased that the paper had generally received favorable and positive evaluations from the reviewer.

Reviewer #3’s comment:

The authors present a calibration method for flowmeters. The special challenges at fuel stations are a wide range of operating parameters and the requirement of using different fluids than hydrogen.

  1. Fig. 8: Was the standard deviation in calculated as uncorrected or corrected standard deviation (divided by N or divided by (N-1))?
  • In the present study, we used corrected standard deviation using dividing by (N-1). As your comments, we also calculated the uncorrected standard deviation by dividing by N. Both results show that the repeatability of measurements are less than ± 0.02%.

  1. Page 8, bottom paragraph: What made the decision to use three measurements for the repeatability test?
  • When we conduct the typical calibration for the flowmeter at the ISO/IEC 17025 accredited laboratories including the national measurement institute in Korea, repeated measurements are performed three times under each flow condition in the entire flow range. That’s why we decided on repeatability test conditions by three measurements.

  1. Based on the results in Figs. 7-12: what is the classification of the flowmeter according to Table 1?
  • Table 1 of OIML R 139-1 described for maximum permissible error of the flowmeter using the hydrogen gas. Therefore, it is difficult to directly compare the results of present the study using the density-matching approach by air and pressure-matching approach by water, but if table 1 is applied, the accuracy class is 2 because the maximum permissible error of the present study is less than 1.5%.

  1. Would a higher number of measurements for the repeatability test change the classification?
  • In the present study, we found that the results of the repeatability tests were quite low and stable in the entire flow range though three repeated measurements were conducted. Therefore, it is expected that there will be no difference in the accuracy classification according to the number of repeatability test.

  1. Reference [5] contains OIML R139-1 in its appendix, which shows a list of requirements, additional to Table 1. Are these requirements all fulfilled?
  • Table 1 and requirements of OIM R139-1 and appendix are described as contents of tests on hydrogen flowmeter installed inside the dispenser of hydrogen refueling stations. In the present study using the density-matching and pressure-matching approach, all possible test conditions were tried to be applied. The on-site test of hydrogen refueling stations for the hydrogen flowmeter according to the OIML R 139-1 will be conducted using Hydrogen Field Test Standard to be developed in the near future.

* Following the comments of reviewer, we carefully checked the spell check in the English language.

We feel that the comments of the reviewer were stimulating and productive. These led to improvements in the revised manuscript. We are appreciative of the reviewer’s efforts for our paper.

Round 2

Reviewer 1 Report

Even if the authors are not able to properly adress the temperature dependency, the new text clearly states that it is for a specific configuration where the temperature of the hydrogen flow meters are stable and around 20°C. I am ok to publish the paper as it is now.

Author Response

2nd Response to Reviewer 1

“Investigation of a Calibration Method of Coriolis Mass Flowmeters by Density and Pressure-Matching Approaches for Hydrogen Refueling Stations” by Woong Kang, Jinwoo Shin, Byungro Yoon, Sunghee Kil, Sangsik Yim, Wonguk Han and Unbong Baek

We are really thankful to the reviewer for his productive comments. We are pleased that the paper had generally received favorable and positive evaluations from the reviewer.

We have prepared a revised manuscript in accordance with the suggestions of the reviewer. As to the specific responses in the revised paper, we would like to note the following modifications. (red letters)

Reviewer #1’s comment:

Even if the authors are not able to properly address the temperature dependency, the new text clearly states that it is for a specific configuration where the temperature of the hydrogen flow meters are stable and around 20°C. I am ok to publish the paper as it is now.

  • As pointed out by the reviewer, in the present study, the operating temperature for the density matching approach was limited to an ambient temperature (around 20 ℃). In the high-pressure gas flow standard system of KRISS, there is the temperature control loop with a capacity 19.7 m3 of for control and stabilization of the operating temperature condition of the test flow meter and reference sonic nozzle as shown in Figure 4.
  • Following the comments of the reviewer, we have added statements for a specific configuration where the temperature of the hydrogen flowmeters with around 20 ℃ in chapter1 Density Matching Approach in High-Pressure Gas Flow Standard System in KRISS (page 5) with red letters.

We feel that the comments of the reviewer were stimulating and productive. These led to improvements in the revised manuscript. We are appreciative of the reviewer’s efforts for our paper.

Author Response

2nd Responses to Reviewer 2

“Investigation of a Calibration Method of Coriolis Mass Flowmeters by Density and Pressure-Matching Approaches for Hydrogen Refueling Stations” by Woong Kang, Jinwoo Shin, Byungro Yoon, Sunghee Kil, Sangsik Yim, Wonguk Han and Unbong Baek

We are really thankful to the reviewer for his productive comments. We are pleased that the paper had generally received favorable and positive evaluations from the reviewer.

We have prepared a revised manuscript in accordance with the suggestions of the reviewer. As to the specific responses in the revised paper, we would like to note the following modifications. (red letters)

Reviewer #2’s comment:

General comment

Please put a space between values and their units: for instance -40 °C, not -40°C; 0.5 g, not 0.5g

  • As pointed out by the reviewer, we put a space between values and their units in the revised manuscript with red letters.
  1. Introduction

Figure 1 is deceptive, as it does not show the pressure ramp regulator. One would falsely assume that the line pressure for the Coriolis meter is 700 bar while this is not true, as this would depend on the mounting position of the Coriolis meter.

  • Figure 1 shows an example of the layout of the hydrogen refueling station in Korea. As pointed out by the reviewer, the pressure ramp regulator is not shown in Figure 1. To avoid misunderstanding from this Figure, we changed the title of the Figure 1 as ‘Example of layout for a hydrogen refueling station and the Coriolis mass flowmeter in the dispenser” with revised manuscripts (with red letters). In addition, we added ‘a pressure ramp regulator’ in the component of the hydrogen refueling station in the revised manuscript with red letters.

The statement about pressure, flowrate and temperature varying widely during a refueling is strictly not correct. Pressure varies but along a ramp, temperature before the cooler is more or less stable and flowrate varies indeed. All these observables depend strongly on the design of the HRS. A figure of such conditions would shed a lot of light on this issue.

  • As pointed out by the reviewer, pressure and temperature condition vary widely during the refueling process. The statement ‘at a pressure 70 MPa and temperature of -40 °C’ was not correct in our manuscript. Following the comments of the reviewer, we revised the manuscript about wide span ranges of pressure and temperature in the revised manuscript with red letters.

Please quote 'The MPE values are 2% to 5%' instead of 2-5%.

  • We have revised the manuscript as red letters in accordance with the comments of the reviewers.

  1. Experimental Methods and Apparatus

2.1 Density Matching Approach in High-Pressure Gas Flow Standard System of KRISS

The SAE J2601 protocol limits max flow rate to 3.6 kg/min.

  • In the present study, four sonic nozzles in the high-pressure gas flow standard system of KRISS were used to test the Coriolis mass flowmeter in the range of flowrate from 1.0 kg/min to 4.5 kg/min. It can cover the maximum flow rate, 3.6 kg/min of the SAE J2601 protocol. To avoid misunderstanding, we clearly revised the manuscript in red letters.

2.2. Pressure Matching Approach in High-Pressure Water Flow Test Facility of KRISS

Typo in the title (presssure instead of pressure)

  • We have revised the manuscript as red letters in accordance with the comments of the reviewers.

  1. Results and Discussion

The Coriolis meter is not described: size, flow rate range,

There is no indication of the zero flow stability of the Coriolis meter.

  • As pointed out by the reviewer, we added the specific information about Coriolis mass flowmeter and the zero-flow stability in the revised manuscript (1 Test Results for Density Matching Approach) with red letters.

In Figures 6 and 7, data show that the Coriolis meter tends to have a negative error with air. The statement that these results are similar to water calibrations is not correct as data from Figures 8 to 11 show. Indeed, Figure 11 shows a clear difference between a calibration with air at 40 bar with respect to a calibration with water at 700 bar. This difference is due to the speed of sound being different in air and in water. Speed of sound in hydrogen at 700 bar is closer to the speed of sound in water than in air. There is very little data available, but this little data seems to show that a water calibration is closer to a hydrogen calibration than an air calibration.

  • As pointed out by the reviewer, two results between the calibration with air at 40 bar and water at 700 bar in Figure 11 show a difference. So, we deleted the sentence for ‘the error between the test results of the density-matching approach and the pressure-matching approach were consistent’. In addition, we have revised the manuscript about the difference in the test results between two approaches due to the speed of sound in accordance with the comments of the reviewers as red letters in 3 Matching approaches and Hydrogen Field Test Standard.

  1. Conclusion

The conclusion about the equivalence of air and water calibrations is not correct. There is a difference as shown in Figure 11.

  • As pointed out by the reviewer, the equivalence of air and water calibrations is not correct. So, we revised delete about the equivalence of two calibration in conclusion and manuscript with red letters.

The conclusion about the methodology being applicable to type evaluation and replacing a hydrogen calibration could potentially lead to a general misunderstanding. There is indeed a general issue with type approval testing of Coriolis meters for hydrogen refuelling applications. The idea that this methodology is the solution is a little over-optimistic. There is at this moment not enough evidence that this is indeed true as no data about equivalence could be shown. It is a pity that the KRISS standard cannot currently be used. Testing Coriolis meters in the lab and then mounting them in a HRS would yield the evidence to prove which method, the density-matching approach or the water calibration, are best suited to reproduce a hydrogen calibration.

  • This is the first step in Korea to investigate the measurement performance capabilities of the Coriolis mass flowmeter with alternative calibration method. We focused to develop the two different matching approaches as potential calibration method and to compare the test results from each high-pressure gas and water flow system. As pointed out by the reviewer, there is difference between two approaches, and we also need the field performance test at the hydrogen refueling station for the Coriolis mass flowmeter tested in calibration laboratory.  
  • So, following the comments of the reviewer, to avoid misunderstanding of the results from present study for readers who consider type approval and replacing hydrogen calibration, we have clearly revised the manuscript with the detailed explanation as red letters in Conclusion

* Following the comments of reviewer, we carefully corrected and revised English language in this manuscript.

We feel that the comments of the reviewer were stimulating and productive. These led to improvements in the revised manuscript. We are appreciative of the reviewer’s efforts for our paper.

We sincerely ask you to reconsider your opinion on the publication of our paper with these response and revised manuscripts.
